# Machine Learning: Using Xception, a Deep Convolutional Neural Network Architecture, to Implement Pectus Excavatum Diagnostic Tool from Frontal-View Chest X-rays

**DOI:** 10.3390/biomedicines11030760

**Published:** 2023-03-02

**Authors:** Yu-Jiun Fan, I-Shiang Tzeng, Yao-Sian Huang, Yuan-Yu Hsu, Bo-Chun Wei, Shuo-Ting Hung, Yeung-Leung Cheng

**Affiliations:** 1Division of Thoracic Surgery, Department of Surgery, Taipei Tzu Chi Hospital, Buddhist Tzu Chi Medical Foundation, New Taipei City 231016, Taiwan; 2Department of Research, Taipei Tzu Chi Hospital, Buddhist Tzu Chi Medical Foundation, New Taipei City 231016, Taiwan; 3Department of Computer Science and Information Engineering, National Changhua University of Education, Changhua City 50074, Taiwan; 4Department of Radiology, Taipei Tzu Chi Hospital, Buddhist Tzu Chi Medical Foundation, New Taipei City 231016, Taiwan; 5Department of R&D, Bamboo Technology Ltd., Taipei City 105037, Taiwan; 6School of Medicine, Tzu Chi University, Hualien 970374, Taiwan

**Keywords:** pectus excavatum, chest X-ray, artificial intelligence, convolutional neural networks, image diagnosis

## Abstract

Pectus excavatum (PE), a chest-wall deformity that can compromise cardiopulmonary function, cannot be detected by a radiologist through frontal chest radiography without a lateral view or chest computed tomography. This study aims to train a convolutional neural network (CNN), a deep learning architecture with powerful image processing ability, for PE screening through frontal chest radiography, which is the most common imaging test in current hospital practice. Posteroanterior-view chest images of PE and normal patients were collected from our hospital to build the database. Among them, 80% were used as the training set used to train the established CNN algorithm, Xception, whereas the remaining 20% were a test set for model performance evaluation. The performance of our diagnostic artificial intelligence model ranged between 0.976–1 under the receiver operating characteristic curve. The test accuracy of the model reached 0.989, and the sensitivity and specificity were 96.66 and 96.64, respectively. Our study is the first to prove that a CNN can be trained as a diagnostic tool for PE using frontal chest X-rays, which is not possible by the human eye. It offers a convenient way to screen potential candidates for the surgical repair of PE, primarily using available image examinations.

## 1. Introduction

Pectus excavatum (PE), also called funnel chest, is a congenital structural deformity of the chest wall. Depression of the anterior chest wall to a certain degree is not merely a cosmetic problem but can also cause exercise intolerance, owing to compromised cardiopulmonary function. PE deformity, with an estimated prevalence of approximately 1 in 300~400 births [1], can cause exercise intolerance owing to cardiopulmonary compression [2]. On examination, the common cardiopulmonary finding is a restrictive pattern in the pulmonary function test (PFT) and decreased right-ventricular function in echocardiography [3].

Pectus excavatum can be corrected by a modified Nuss procedure, a minimally invasive method, where stainless-steel bridge metal bars are placed retrosternally to strut the depressed chest wall [4] Although surgical repair has been proven to improve cardiopulmonary function in patients with PE [5], there is much to be regretted considering some PE patients miss the opportunity for surgical correction at the optimal age due to a delayed diagnosis of the disease. Patients may feel embarrassed about their appearance and may not be willing to talk about it. Some patients may not even be aware of its existence, and therefore they cannot ascribe their exercise intolerance and cardiopulmonary symptoms to PE, making the accessibility of PE diagnosis more important. Clinically, the image diagnosis of PE still relies on manual measurement of the thoracic diameter on chest computed tomotraphy (CT). Physicians measure the maximum transverse diameter of the thoracic cage divided by the shortest distance from the sternal to the anterior vertebral body to get the Haller index, which can indicate the severity of pectus excavatum. The cut-off value of Haller index 3.25 has been widely used for pectus excavatum diagnosis for decades based on the preliminary report of Haller et al., which revealed that all patients who received operative repair had a Haller index greater than 3.25 (Figure 1) [6]. However, chest CTs are not economical and are not frequently requested imaging test compare to frontal view chest X-rays during routine health examination. Moreover, a chest CT examination takes 30~170 times more radiation dose exposure than a chest X-ray [7]. Since a human can hardly differentiate pectus excavatum from a frontal view chest X-ray, our proposal is to use artificial intelligence to accomplish this task. Our aim is to train a machine learning model to diagnose pectus excavatum from plain frontal chest radiography automatically, without the need for a chest CT. It would not only economize manual labor and medical expenses as a screening tool, but would also ensure that the potential diseased candidates could be screened out for surgical repair to improve their quality of life, while helping them to avoid extra radiation exposure.

Of note is that the PE diagnosis is unlikely from frontal chest radiograph images without a chest lateral view or CT images.

### Convolutional Neural Networks (CNN) and Medical Image Detection

Machine learning and artificial intelligence in the computing field aims to develop algorithms to teach computers to detect patterns in data autonomously. Artificial neural network (ANN) architecture and the further derived convolutional neural network (CNN) architecture, as computing methods, can be used to develop an algorithm to achieve this purpose. CNNs can be very good feature extractors which use filters to obtain convolutional layer output, thereby reducing training parameters while keeping the accuracy. We can also repeat the convolutional layers several times to go deep to make the model match the data pattern more precisely [8]. Starting with LeNet-style [9], the following refined well-known CNN architectures, VGG-style network, Inception and Inception-ResNet have proven their powerful performance in image classification tasks. Some researchers have applied these models for the interpretation of chest X-ray images to detect pneumothorax, nodules, masses, opacity and fracture [10,11,12]. In recent research, the authers used VGG16 and VGG19 to identify pectus excavatum from chest computed tomography images [13]. However, artificial intelligence has not been yet applied for detecting PE from frontal chest X-rays. Among the above CNN models, Xception is the latest proposed model derived from Inception which has better performance on the ImageNet dataset and JFT (internal datasets used at Google) dataset. Xception also had a smaller parameter count and higher training speed compared to Inception V3 [14]. For reasons outlined above, we adopted Xception as our model training algorithm.

## 2. Materials and Methods

All the radiographs and reports in our study database were obtained from the Taipei Tzu-Chi Hospital, Taipei, Taiwan, ROC, and were fully anonymized. The study was approved by the Ethics Committee and Institutional Review Board (IRB No: 11-XD-109). The requirement for patient consent was waived by the Institutional Review Board.

Images of the posteroanterior-view chest radiographs for model implementation were retrieved in JPEG format with 1760 × 2140 pixels from the clinical picture archiving and communication system (PACS). These images were downsized to a resolution of 224 × 224 pixels and matted through YOLOv4, making them suitable for image recognition by a CNN [15]. Every now and then, some extra mark can be seen for clinical use for peripheral chest X-ray images. By object detection (YOLOv4) identifying the chest cage, we can exclude those unrelated parts, such as peripheral marks usually used for clinical label purposes and the limb girdle. Now that the CNN model can extract the diseased features, the chest cavity is used for model training (Figure 2).

### 2.1. Data Set

Images were captured from 1 January 2006 to 31 December 2020 from patients aged 12–50. The collected images were divided into two independent datasets: CXRs from pectus excavatum patients (PE) and normal (N). The PE dataset included images from patients whose Haller index was determined to be greater than 3.25, measured and calculated manually through chest CT, all of whom were diagnosed with PE and underwent surgical treatment in our hospital. The images in the N dataset were obtained from a normal group with no abnormalities in either frontal chest X-ray or chest CT. All the patients in the normal group were verified to have no PE, based on the Haller index less than 3.25 calculated manually from their chest CT images. The images in the two datasets were randomly split into an 80:20 ratio for the training and test sets, respectively, by a computer program. The test set, which is a holdout dataset, was never encountered by the algorithm during training. This set was used to evaluate the trained models.

### 2.2. Model Development

The convolutional neural network architecture used for our model implementation was Keras Xception (version 2.5.0) in Tensorflow (2.5.0) [11]. This deep learning task was executed in Google Colab using up to Tesla P100-PCIE. In the normalization process, we normalized the X-ray intensity value of the 224 × 224 pixel image in the 0~255 range as input data, computing using the process:(1)$$[(x_{i}\;-\;\min\;(x))/(\max\;(x)\;-\;\min\;(x))]\;\times \;255$$

The convolutional layers of Xception architectures were completely unchanged. After going through transfer learning, the Xception architectures used for our model development has its retained trained weights because it has been pretrained with an ImageNet large-scale, multi-class labels dataset. The process flow of our model algorithm is exhibited in Figure 3. We applied a cross-entropy loss function to obtain the final output of the sigmoid function and Softmax classifier with two outputs to obtain our binary final result. Nadamax was used as the optimizer. The batch size was set as 32. The learning rate and learning rate schedule were all set to their default values. We did not use dropout or image augmentation for our model because we have sufficient image data to compare to pre-existing pectus excavatum image evaluation studies. We set early stopping to avoid overfitting. The hyperparameters used during model development are listed in Table 1.

### 2.3. Model Evaluation

The primary outcome is the model’s performance in distinguishing patients with PE from the frontal chest X-rays on the test set. The performance is depicted as a confusion matrix. The accuracy, sensitivity, specificity and positive predictive values were computed. The receiver operating characteristic (ROC) curves were plotted using matplotlib (version 3.2.2).

### 2.4. Statistical Analyses

SPSS Statistics for Windows, version 24 (IBM Corp., Armonk, NY, USA), was used for statistical analyses. The investigated parameters in our population were normally distributed using the Kolmogorov–Smirnov test. Continuous data are depicted as mean ± standard deviation, whereas categorical data are depicted as a count (%). The patient characteristics of the PE and N datasets were compared through student’s t tests.

## 3. Results

A total of 2027 posteroanterior chest radiograph images were utilized, with 774 images in the PE dataset from 520 patients and 1253 images in the N dataset (normal group) from 667 people. The PE group comprised 84.6% men and 15.4% women with an average examined age of 23.4 ± 7.8 years. The N group comprised 49.2% men and 50.8% women with an average examined age of 41.0 ± 6.7 years. A total of 27.3% of patients in the PE group and 35.8% in the N group exhibited more than one image because they underwent a series of chest X-ray examinations. We split our dataset into 80% training and 20% test sets. The epidemiological data in the PE group had a significantly higher Haller index (4 ± 1.2 vs. 2.5 ± 0.37) and the mean age was younger than that of the N group (23.4 ± 7.8 vs. 41.0 ± 6.7). In our PE group, the proportion of men was dominant (84.6%); nevertheless, women had a higher Haller index (*p* < 0.001). The mean Haller index of women was higher in both the PE and N groups (*p* < 0.001). The prevalence of obvious scoliosis with Cobb’s angle more than 20°, noted concomitantly during chest X-ray review, was 9.6% in the PE dataset and 4.2% in the N dataset. The shapes of chest wall depression of PE were not identical in our patient group—53.1% were asymmetric (Table 2).

### Model Performance

As previously mentioned, our model was implemented using Xception, a standard network architecture in the Keras deep learning library [14]. We set early stopping in our script code, wherein the training process is stopped if there is no improvement in accuracy after three epochs. The highest accuracy for the model, evaluated on the test set, was realized after 28 epochs on the training set. The test set used for evaluating the performance of our model was not used previously during the training process. The confusion matrix indicating our model performance is shown in Figure 3, with an accuracy of 0.973 ± 0.005, precision of 0.986, recall of 0.943, F1-score of 0.964 and AUC of 0.976 ± 0.014 (Table 3; Figure 4) [16].

## 4. Discussion

The development of convolutional neural networks (CNN) drives the progress of imaging diagnostic medicine. In addition to automatic detection [17], it is also possible to identify image features that are invisible to the human eye [18].

Since the release of the NIH ChestX-ray14 dataset [19], which contains 112,120 labeled frontal chest X-rays, artificial intelligence algorithms for automated diagnosis from chest radiographs have been developed. In addition to the ChestX-ray 14 (originally ChestX-ray 8), there are several large open-access chest X-ray datasets worldwide that may be utilized for future work, such as CheXpert from Stanford [20], MIMIC-CXR from M.I.T [21] and the well-known Alicante hospital chest X-ray datasets with large chest X-ray image data [22]. These datasets enable the CNN architectures to be trained as chest X-ray automated diagnostic models. The most popular among these chest X-ray autodetection algorithms is CheXNet, which can replicate radiologist-level pneumonia detection [23]. The trained neural network CheXNeXt can concurrently [12] detect 14 different pathologic findings, as classified in the NIH ChestX-ray 14 dataset on frontal-view chest radiographs. Many such algorithms have proven to be just as feasible, valid and competitive as certified radiologists for certain diagnoses [24,25]. Notably, none of these databases include the PE category. The 14 pathology classes of ChestX-ray 14, the largest and most well-known chest X-ray database, are pneumonia, pneumothorax, consolidation, atelectasis, nodule, mass, infiltration, cardiomegaly, emphysema, edema, effusion, fibrosis, hernia and pleural thickening. However, this does not include PE. To our knowledge, no attempt has been made to diagnose PE from frontal chest radiographs. Previous studies have focused on diseases which require label annotation from radiologists on chest X-rays compared to the result of machine reading [11,12,26]. Our study applies the CNN algorithm innovatively to detect PE; the distinguishing features of which are beyond the resolving power of human eyes. There is no need for manual label annotation in our study. Nevertheless, in real world clinical practice, the radiologist is not obligated (or unable) to diagnose PE from frontal chest X-rays. The results of our study reveal that training established CNN architecture can distinguish PE on frontal chest X-rays, which is not possible by the human eye.

For model implementation, we selected Xception, a novel deep CNN architecture that replaces depth-wise separable convolutions in the inception algorithm. Xception performs better than Inception V3 on the JFT dataset (an internal Google dataset) [14]. Some studies compare several CNN architectures, such as VGG, Inception, Xception, ResNet, DenseNet and EfficientNet to evaluate the machine learning performance and parameter optimization of frontal chest X-ray interpretation [26,27]. Taylor et al. used VGG16, VGG19, Inception, Xception and ResNet, and manipulated their parameters to determine the best model for detecting pneumothorax in frontal chest X-rays. The best performing models published for the prediction of pneumothorax have a validation AUC of 0.94. The performances of these models are approximated after hyperparameter optimization [14]. In our study, we have reported that Xception is a promising CNN architecture for the detection of pneumothorax after parameter optimization, as it has a high accuracy of 0.973 and an AUC of 0.976. Further studies are required to train different CNN architectures for better model establishment.

As PE is a low-prevalence disease, the available data are limited. At the beginning of model development, we split the database into 70%, 15% and 15% for the training, validation and test sets, respectively. Generally, datasets are often split into training set, validation set and test set. However, we could not train the model successfully on this setting because of the limited number of images. After discarding the validation set and redistributing the data into training/test sets to increase the data in both sets, the training model could be established. K-fold cross-validation is also a modified validation method to split the training set into K parts which can be used as a validation set alternatively. K-fold cross-validation is also a good way to evaluate the training model and tuning to optimize the hyperparameter when the training data are limited [28]. It can be adopted for our further experiment compared with the previous model. We selected adolescent and adult patients for model implementation for two reasons: (i) the best time for PE repair surgery is when patients are in their adolescence or young adulthood [29,30]; (ii) chest X-rays confirmed as normal or with PE through chest CT during childhood are rare, and those available could be extreme data that may disturb model training.

Considering the limited data for model training, the CNN architecture used for model implementation was pretrained on the ImageNet dataset in the Keras library [31]. ImageNet is an image database with more than 14 million images classified into more than 20,000 categories with hand-annotated objects on the pictures. Ke et al. (2021) compared the transfer performance and parameter efficiency of 16 popular CNN architectures on a large chest X-ray dataset, and found that ImageNet pretraining provided a statistically significant improvement in the algorithm performance [27].

### Limitation

The population of our study is all Asian and male-dominant and is limited in comparison to those of other deep machine learning studies. We require more diverse pooling data that include other races and more female data. This would enhance the validity of the machine model and aid further study to compare different subgroups. In addition, evaluation datasets from other institutions for external validation are requested. As the Haller index is extensively used to quantify PE, and to indicate and evaluate the results of surgery [6], we used a Haller index of 3.25 as the cutoff to separate the data into normal and disease groups for deep learning training in our study. However, the Haller index is not necessarily related to the severity of the patient’s symptoms and cardiopulmonary function. Further studies can combine the relative symptoms and other parameters, in addition to chest X-ray images, to render the proposed diagnostic model more effective. Additionally, the image processing in our study was just downsizing and simple intensity value normalization. In fact, when processing large amounts of image information, there are state-of-the art methods proposed to help in denoising unnecessary signals as well as in preserving the image texture and details. Wavelet denoising or denoising-compressed sensing by regularization (DCSR) can be applied for image processing [32,33]. By the denoising method, feature simplification may improve overfitting problem in our machine learning model, which can be verified by further work.

## Figures and Tables

**Figure 1 biomedicines-11-00760-f001:**
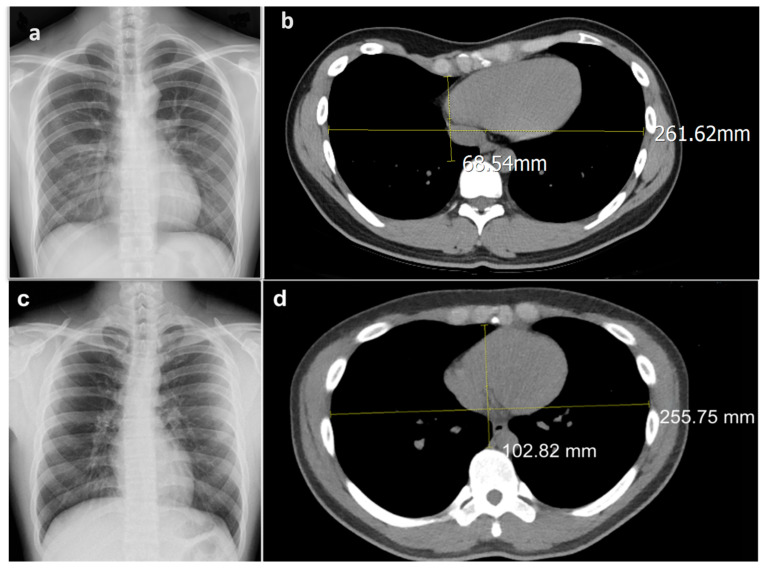
Pectus excavatum is diagnosed by Haller index > 3.25. Calculation of Haller index by measurement of SVD (sternual-vertebral distance) divided by TD (transverse chest diameter) from chest computed tomography (CT). Sample images used in our study: (**a**) posteroanterior-view chest X-ray of a 24-year-old patient with PE with (**b**) CT calculated Haller index of 3.82. (**c**) Posteroanterior-view chest X-ray of a 24-year-old patient with no abnormal radiographic finding with (**d**) CT calculated Haller index of 2.49.

**Figure 2 biomedicines-11-00760-f002:**
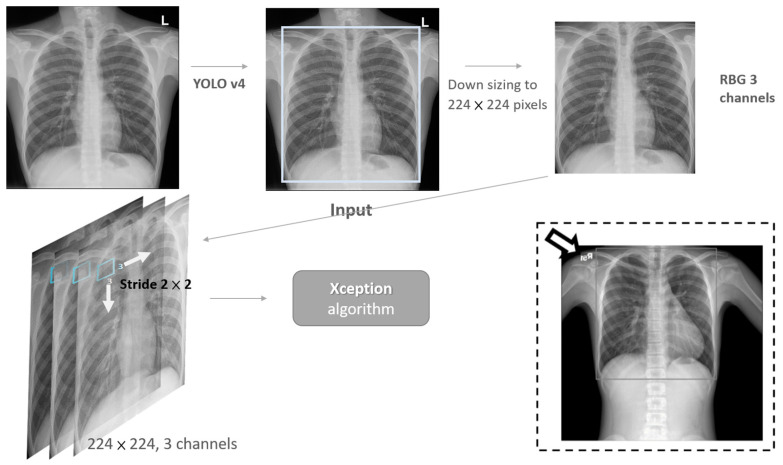
Image processing before input to Xception architecture. Miniature in dashed box shows how images were matted through YOLOv4 processing. The frame was just abutting the chest cage to exclude unnecessary features. Arrow: extra mark in clinical use.

**Figure 3 biomedicines-11-00760-f003:**
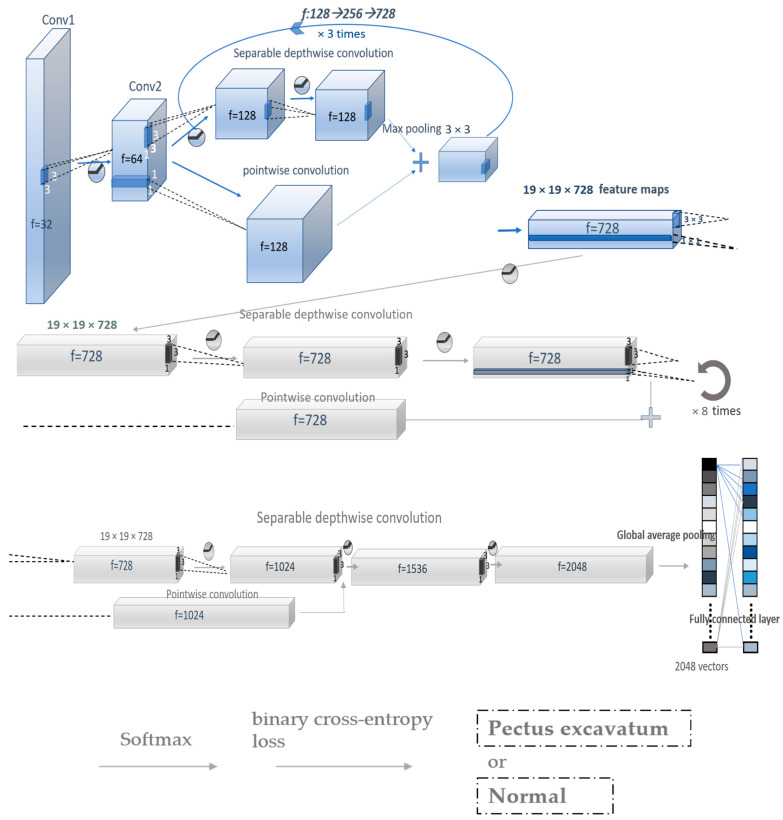
Xception architecture diagram: Xception is a complex multiple layers (deep) CNN structure which contains both separatable depth-wise and pointwise convolutional layers. The Xception architecture is composed of entry flow, middle flow and exit flow. After going through the entry flow (blue part), the data then go through the middle flow which is repeated eight times (middle grey part), and finally through the exit flow. After fully connected layer, all vectors go through SoftMax function then binary cross-entropy loss to obtain binary classification results.

**Figure 4 biomedicines-11-00760-f004:**
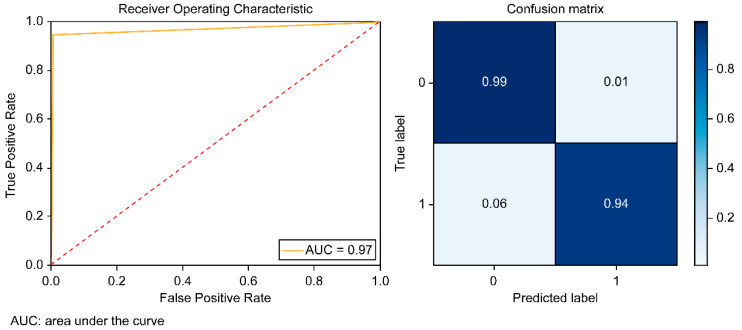
Diagnostic performance of the model on the test set: ROC curve (**left**) and confusion matrix (**right**).

**Table 1 biomedicines-11-00760-t001:** Hyperparameters during model training.

Parameter	Values	Explanation
Arch	Xception	Architecture: Xception
Imgshape	224 × 224	Image shape: downsized image pixels
pooling	Global average	Pooling method after convoluted filter layers
LR	default	Learning rate
LR schedure	default	changes the learning rate during learning
Batch size	32	a number of samples processed everytime the model is updated
dropout	0	Dropout setting applied to fully connected layers
Augmentation (zoom, shear, rotation)	(0, 0, 0)	Image transformation to expand data size
optimizer	nadamax	Optimization algorithm used for training
Batch normalization	no	A layer inserted before the pooling layer

**Table 2 biomedicines-11-00760-t002:** Patient characteristics of the Training and Test Data Sets.

Characteristic	PE Data Set	N Data Set	*p* Value
Total No. of chest X-rays Mean examined age (y)	774 23.4 ± 7.8	1253 41.0 ± 6.7	<0.001
Patients (n) Men (n) Women (n)	520 440 (84.6%) 80 (15.4%)	667 328 (49.2%) 339 (50.8%)	
Haller index, mean ± SD Men Women	4 ± 1.2 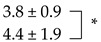	2.5 ± 0.37 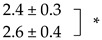	<0.001
Patients with (n) 1 chest X-ray 2 chest X-rays ≥3 chest X-rays	378 (72.7%) 125(24.0%) 17 (3.3%)	428 (64.2%) 102(15.2%) 137(20.6%)	
PE shape (n) symmetric Asymmetric Right site depression Left site depression	244 (46.9%) 276 (53.1%) 176 (33.8%) 100 (19.3%)	100 (100%) NA NA NA	
Scoliosis (n) ^#^	50 (9.6%)	28 (4.2 %)	

No. n: number; y: year-old; PE: pectus excavatum; * *p* < 0.001; ^#^ Cobb’s angle > 20°.

**Table 3 biomedicines-11-00760-t003:** Detection performance of PE from frontal chest X-rays.

Accuracy (95% CI)	Precision	Recall	F1-Score	AUCOC (95% CI)
0.973 (0.968–0.978)	0.986	0.943	0.964	0.976 (0.962–0.990)

CI: confidence intervals; AUROC: area under the receiver operating characteristic curve.

## Data Availability

Not applicable.

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
