# Peer review of "Machine Learning: Using Xception, a Deep Convolutional Neural Network Architecture, to Implement Pectus Excavatum Diagnostic Tool from Frontal-View Chest X-rays"

_biomedicines, 2023, doi:10.3390/biomedicines11030760_

Round 1

Reviewer 1 Report

The manuscript describes the experimental results of a CNN model applying for detecting Pectus Excavatum with Chest X rays. 

The minor problems, I think , are followed.

1.     The second paragraph of the introduction (Starting from line # 40 to 50) is not needed in this manuscript. The point of the paragraph is not related to this manuscript.

2.     You should describe the default values used in the experiments(line #114).

3. When experimenting, all used values including default in the system should be described in detail for other researcher's experimnemts.

4.  You used the data which were already categorized by a traditional criteria (Haller index).   As described in the conclusion by you, it my be better decide the boundary by learning of the model with the data which were divided by experters.

Author Response

  1. Thank you for your suggestion. We deleted the irrelevant paragraph just like you mention (Marked version: L.51~71; L.145~155). We did major revision to our introduction .We also added CNN introduction part which was related to our work to the second paragraph of our introduction.
  2. &
  3. Thank you for the recommendations. We revised our methodology part of our article (Mared version: 2.2 Model development). The default value and the algorithm flow are described more detail on this part. We also added table 1. to sum up our default value.
  4. Thank you for the comment. The traditinal criteria of pectus excavatum disagnosis require experters (physicians, operators, radiologist...) to measure the value on the chest CT and calculate the Haller index. It is true that it would be better to let machine to decide the boundary. However, we have to train the model first by teaching the abnormal CXR from normal CXR first. Therefore, we used pre-categorized dataset. To optimize the model, we sure should let machine learning adjust the boundary based on our existing model in future work.

Reviewer 2 Report

In this study, the authors proposed a CNN architecture for Pectus Excavatum Diagnosis from Frontal-View Chest X-Rays. Although a few promising results have been achieved, some major points should be addressed as follows:

1. Data processing (i.e., standardization, normalization, augmentation, etc.) is important for such kind of study to have a good prediction model. However, the authors skipped these steps. It is suggested to add.

2. Why did the authors only consider frontal view in their models?

3. There are many pre-trained models released, why did the authors only consider Xception in their implementation? How to know Xception worked well in this problem rather than other models? At least, the authors should have some baseline comparisons to explain their choice.

4. The authors should compare the predictive performance to previously published works on the same problem/data.

5. Model interpretation (i.e., using Grad-CAM) should be added to see how the models learned the data/features.

6. There must have an external validation to evaluate the performance of the models and see the generalization of the models.

7. Deep learning or CNN is well-known and has been used in previous biomedical studies i.e., PMID: 34915158, PMID: 35648374. Thus, the authors are suggested to refer to more works in this description to attract a broader readership.

8. Quality of figures should be improved.

9. The authors are suggested to conduct cross-validation in their training process.

10. Source codes should be provided for replicating the study.

11. The use of "deep machine learning" in the title looks confusing. It is suggested to change it.

Author Response

  1. Thank you for your kind suggestion. We revised our methodology part and describe our algorithm flow, data processing more detail (see 2.2 model development) which include how standardization, normalization, augmentation, batch size... of our data processing. We also sum up the information in table. 1 and Figure.2                                                                 
  2. chest CTs are not an ideal tool but frontal chest X-rays for pectus excavatum surveillance. Frontal view chest X-rays are more economical and frequently requested imaging examination during routine health examination. Besides, chest CT examination takes 30~170 times of radiation dose exposure more than chest X-ray. (Line. 88~98)                       
  3. Thank you for your recommendation, based on which, we added second paragraph of our introduction to describe the evolution of CNN and why we choose Xception as our architecture for model training (L.122~143). It was sure that we did not know how the performance of Xception at the beginning, but it is more complex and latest released by Keras that's why we choose it, and result comes out well. We will working on the further comparison of different model performance in our future experiments.        
  4. Thank you for your advice.  We added comparison context on line.134~137 to compare our work to previously published work. Both works are using AI to detect pectus excavatum image. Among previously published work, we are the first one trying to use AI to diagnose pectus excavatum on frontal chest X-ray image.                                                                                
  5. We appreciate your comments and suggestions. Your proposal is well taken.  Grad-CAM model interpretation is a good way to present how machine learns the features of the images. Please allow us the second revision opportunity. We will add Grad-CAM figure on published version.     
  6. Thank you for your advice.  For readers to understand that Testing the original prediction model on a group of fresh patients to see if it performs as expected is known as external validation. In this study, we split 20% of our dataset into test group (Line 181~185). The images in test group has never seen by model during training. It is much like internal validation. Unfortunately, we may not obtain another data for external validation. This study only conducted for a single center experience. No images data from outer institution obtained at present.  We also added this concern as a limitation.                                                                                                         
  7. We appreciated the reviewer's kind suggestion and advice. The two extraordinary articles gave us more mindstrom on AI application in image medical diagnostic field. We think it will be a pleasure if more readers know the information. We added the reference at the beginning of our disccussion paragraph (Line. 286~288)                                                            
  8. Reviewer's concern has been well taken. We refined our figure. We deleted our previous Figure 2.. The new Figure 2. contained more detail of the model information as well as algorithm flow has been faithfully represented.                                                                                                     
  9. We appreciated your suggestions. We tried to seperate our data into three subset (train, validation, test 70/15/15) as other AI models do. However, the model can not be trained successfully in this setting. The model can be trained successfully when we slipt the dataset into train (80%) and test (20%) set which mau due to more available data for training set. However, based on your suggestion, we found K-fold cross validation may be a good solution of this problem, which is worth as a method when we refind or build our model next time. We also added the relative description on discussion paragraph (Line.332~342)                                                              
  10. Reviewer's suggestion has been well taken. We will upload our model coding as appendix A.                                                                                      
  11. Thanks for your friendly reminder. We modified our title as more straight forward and correct way. "Machine Learning: Use Xception, a Deep Convolutional Neural Network Architecture, to implement Pectus Excavatum Diagnostic tool from Frontal-View Chest X-Rays"

Reviewer 3 Report

This study is very interesting but the manuscript lacks substance and form.

In order to improve their manuscript, the authors must comply with our recommendations:

1. The authors should explain in a clear and detailed way the conditions of use of Yolov4.

2. The authors should at least recall very efficient methods to reduce unnecessary noise such as denoising in a simple and quasi-optimal way by wavelets [1] or during acquisition based on the concept of compressed sensing [2].

1] doi: 10.1109/WoSSPA.2013.6602330.

[2] https://doi.org/10.3390/s22062199

3. The authors should analyze the following reference [3] and perform a comparison.

[3] doi: 10.1038/s41598-020-77361-y

4. Perform a Haller index swot study

5. How is p-value calculated and what test is this calculation based on?

6. Some incorrect notations and typos need to be corrected as for example :

- In line 101, it is incorrect to write "(<3.25)"

- In line 140, it is incorrect to write "(<0.001)"

Author Response

  1. Thank you for your suggestion. We added more detailcontext about YOLOv4 in the Materials and Methods paragraph (Line. 165~169). "Every now and then, some extra mark can be seen for clinical use at the peripheral of chest X-ray images. By object detection (YOLOv4) identifying the chest cage, we can exclude those unrelated part such as perpheral marks usually used for clinical label purpose and the limb girdle. Now that CNN model can extract the diseased features, the chest cavity, for model training(Figure 2)."                                                                                              
  2. We appreciated your suggestions. Your opinion broad our mind in image denoising field. It is worth as a method that we can refind our present model or use it to build our model next time. We also added this important information in our limitation paragraph (Line 364~370).                                 
  3. Reviewer's suggestion has been well taken. We compare the article which was also a pectus excavatum diagnosis tool build on CNN architecture. " In recent research, the authers used VGG16 and VGG19 to identify pectus excavatum from chest computed tomography images[13]. However, artificial intelligence has not been yet applied for detecting PE from frontal chest X-rays." (Line 134~137).                                                                          
  4. Thank you for your suggestion. Haller index as a diagnostic index of pectus excavatum has its strengths and weaknesses, there is also an opportunity to find a more efficient or more precise diagnostic method .Due to the space of the paragraph, we did SWOT analysis by discuss in context rather than a SWOT analysis chart.  The SWOT analysis has been described in the context (Line. 81~89 “Clinically, the image diagnosis of PE still relied on manually measurement of thoracic diameter on chest computed tomography (CT). Physicians measure maximum transverse diameter of the thoracic cage divided by the shortest distance from the sternal to the anterior vertebral body to get Haller index which can indicate the severity of pectus excavatum. The cut-off value of Haller index 3.25 has been widely used for pectus excavatum diagnosis for decades based on the  preliminary report of Haller et al., which revealed that all patients received operative repair had a Haller index greater than 3.25 (Figure 1.)[6]. -However, chest CTs are not an ideal tool but frontal chest X-rays for pectus excavatum surveillance.”) (Line 357~363 “As the Haller index is extensively used to quantify PE, and indicate and evaluate the results of surgery [6], we used a Haller index of 3.25 as the cutoff to separate the data into normal and disease groups for deep learning training in our study. However, the Haller index is not necessarily related to the severity of the patient’s symptoms and cardiopulmonary function. Further studies can combine the relative symptoms and other parameters, in addition to chest X-ray images, to render the proposed diagnostic model more effective.”)                                
  5. We thank the reviewer’s comments and suggestions. We examined the check of normality for variable used Kolmogorov–Smirnov test. All variables pass normality test. Next, to test mean difference for PE and N datasets used independent samples t test (i.e., p-value calculated by such test). Independent samples t test also used for testing mean difference (of Haller index) among women and men groups in Table 1 (i.e., p-value calculated by such test). For the AUC, a point estimate of the AUC of used the Mann-Whitney U estimator (i.e., p-value calculated by such test). The confidence interval for AUC used the Wald Z statistics.          https://analyse-it.com/docs/user-guide/diagnostic-performance/auc           

  6. Thank you for your friendly reminder. We've corrected those incorrect notations and typos.

Round 2

Reviewer 2 Report

My previous comments have been addressed well.

Author Response

Thank you very much for your valuable comments.

Reviewer 3 Report

The manuscript has been improved. It can be published almost as is, provided that the following corrections are made:

- correct the multiplied sign in the relation in line 195: Indeed, instead of putting "*", the mathematical symbol for multiplication "x" should be put.

- redraw figure 2 B and make it more readable.

Author Response

(The authors gave the same response as above.)
